# Public Knowledge about Dementia in China: A National WeChat-Based Survey

**DOI:** 10.3390/ijerph16214231

**Published:** 2019-10-31

**Authors:** Dan Liu, Guirong Cheng, Lina An, Xuguang Gan, Yulian Wu, Bo Zhang, Sheng Hu, Yan Zeng, Liang Wu

**Affiliations:** 1Brain and Cognition Research Institute, Wuhan University of Science and Technology, Wuhan 430081, China; liudan125@wust.edu.cn (D.L.); chengguirong@wust.edu.cn (G.C.); anlina@wust.edu.cn (L.A.); xuguanggan@wust.edu.cn (X.G.); Wuyulian@wust.edu.cn (Y.W.); wustzhangbo@wust.edu.cn (B.Z.); zengyan68@wust.edu.cn (Y.Z.); 2Big Data Science and Technology Institute, Wuhan University of Science and Technology, Wuhan 430081, China; 3School of Information Engineering, China University of Geosciences, Wuhan 430074, China; husheng@cug.edu.cn; 4National Engineering Research Center for GIS, Wuhan 430074, China

**Keywords:** dementia, knowledge, risk factors, WeChat

## Abstract

Dementia is a serious public health problem. The more extensive dementia knowledge is, the more conducive it is to early prevention and treatment of dementia. However, no assessment of the general population’s dementia awareness has been conducted so far in China. Thus, this study assessed the national public knowledge of dementia based on mobile internet in China. We assessed 10,562 national respondents recruited based on the most popular social networking service in China, WeChat and analyzed the data using quantitative methods. The overall correct rate of total dementia knowledge was 63.14%. Only half of the participants (50.84%) could identify risk factors accurately. The level of dementia knowledge was positively associated with high education, city residency, and experience of exposure to information on dementia. The sandwich generation (aged 20–60 years) had the highest level of dementia knowledge. Chinese people were found to have a low level of knowledge about dementia, especially those aged over 60 years, with low education and living in rural areas. Further educational programs and campaigns are needed to improve dementia knowledge, with greater focus on the older population as the target audience, emphasis on dementia risk factors as educational content, correcting misconceptions about dementia, and providing more experience of exposure to dementia.

## 1. Introduction

Dementia is a chronic disease characterized by a progressive and irreversible decline in cognitive functions, which mainly affects older people, particularly those aged over 65 years. Alzheimer’s disease (AD) is the most common type (60–80% of cases) of dementia. Today, with the increasing population aging in all parts of the world, the number of newly diagnosed dementia cases is over 9.9 million each year worldwide, equating to 1 new case every 3.2 seconds. Approximately 58% of dementia patients live in low- and middle-income countries, and this percentage is expected to rise to 68% by 2050 [1]. In China, the prevalence dementia is estimated at 6.19% [1] , and 62.8% of patients with dementia have AD [2]. The number of patients with dementia in China accounts for approximately 25% of the entire worldwide population with dementia [3]. AD is one of the costliest chronic diseases for society [4], creates a heavy burden on both patients’ families and society, and poses a huge challenge to economic development and elderly healthcare. The total cost of dementia, including direct, indirect, and intangible costs, was calculated as US $957.56 billion in 2015 and is expected to reach US $2.54 trillion by 2030 and US $9.12 trillion by 2050 [5]; these costs are significantly greater than what was previously predicted [6]. China accounts for a high proportion of the total worldwide costs of dementia, accounting for 17.52% of the cost in 2015, and is predicted to account for 18.71%, 20.00%, 20.80%, and 20.70% in 2020, 2030, 2040, and 2050, respectively. The total cost of AD in China in 2015 represented 1.47% of the Gross Domestic Product (GDP), which is higher than the global average (1.09%) [5]. 

Alzheimer’s Disease International (ADI) has recommended that every country should have a national plan to deal with this major public health crisis [6]. A previous study found that greater dementia knowledge by the general public was associated with higher levels of competence in recognizing dementia-related behaviors [7]. Lack of knowledge often leads to problematic delays between symptom onset and healthcare provision [8,9]. The World Health Organization (WHO) Global Action Plan on Dementia encourages all member countries to implement campaigns to raise public and professional awareness about dementia and to promote the creation of dementia-inclusive societies [6]. Prior to the implementation of large-scale education campaigns, it is vital that we carefully elucidate the current state of public knowledge in order to guide the design of interventions that maximize limited financial resources.

Much of the focus of research on knowledge of dementia has been on community-dwelling older adults [10]; family caregivers for patients with AD [11]; mental health providers [12]; psychologists [13]; medical school students [14]; and outpatients [15], rather than the general public. Recently, some studies investigating the public dementia knowledge have mainly been conducted among developed countries in Europe, America, and Asia [9,10,16,17,18,19]. Differences in the level of dementia knowledge, however, may derive from ethnicity, nationality, and culture [20,21].

In China, several studies exploring the dementia-related knowledge of different groups, such as community residents [22,23], family and hospital caregivers [12,24], and nursing assistants [25] have been conducted. Different groups of people have different levels of dementia knowledge, which are associated with region, education, experience of dementia. Other scholars have conducted an extensive survey in five representative cities of China, focusing on the fear of AD [26]. Therefore, a survey with a larger sample size and a wider population coverage is required to reflect the actual knowledge levels of Chinese people about dementia. The present study through WeChat fiend circles was aimed at elucidating the extent of the Chinese general public’s knowledge about dementia using a larger sample than used in previous studies. To the best of our knowledge, this is the first time that a national public survey has been conducted on dementia knowledge based on the mobile internet and WeChat friend circles in China.

## 2. Materials and Methods

### 2.1. Participants

We recruited 238 second-year students, who came from all over our country, from our medical school as seed investigators and trained them to conduct the survey through WeChat. Participants who met the following criteria were included: (1) aged 18 years and older, (2) can read a Chinese questionnaire, (3) are WeChat users, (4) volunteered for the survey, (5) could submit survey responses using the same IP address only once. The exclusion criterion was being unable to understand the questionnaire. First, people participated in the survey by scanning our seed investigators’ WeChat Moments and then they circulated the surveys in their WeChat friend circles. We obtained an overall sample size of 10,562 participants. Prior to conducting the data analysis, we excluded all the respondents who provided incomplete responses to demographic items. Overall, we excluded 762 (7.2%) of the initial respondents, leaving 9,800 cases for analysis. The current study was approved by the ethics committee of Wuhan University of Science and Technology (No. 20180143). Only with the informed consents of the respondents did the questionnaire begin, and the survey was conducted anonymously in order to protect the respondents’ privacy.

### 2.2. Measures and Procedure

Several existing tests assess knowledge about dementia and AD [9,16,27]. The Alzheimer’s Disease Knowledge Scale (ADKS) developed by Carpenter et al. [27] has been widely used to assess knowledge about dementia and AD [28,29]. There is a Chinese version of the ADKS scale translated by professor Yu Hong-Mei, which is considered to have good reliability and validity but has not been widely used [30]. Because of Chinese socio-economic factors, personal car driving is not common, so we removed the items related to safety driving. Second, previous research has revealed that 84.9% of AD patients are cared for by non-professional caregivers at home [31]. On the one hand, China undeniably faces a shortage of nursing homes, and on the other hand, the Chinese tradition of filial piety mandates that the elderly should be cared for by their families when they lose the ability to live independently [32]. Recent studies have found that cognitive stimulation therapy [33] and cognitive stimulation activities [34] can improve cognitive function in dementia, and engaging in mentally stimulating activities (reading books, computer use, social activities, playing games, craft activities), particularly in late life is associated with a decreased risk of MCI among community-dwelling old persons [35]. Meanwhile, Chinese people always refer to AD as “senile dementia”. Considering that the general public may not clearly distinguish between ‘‘dementia’’ and ‘‘Alzheimer’s disease’’ [16], these terms were used synonymously in the national public survey. Thus, based on the Chinese version of ADKS, we made a comprehensive adjustment to the questionnaire’s contents to suit the national survey of the Chinese population. 

The modified dementia knowledge scale contained 27 true/false items covering different domains: symptoms (8 items), risk factors (8 items), life impact (4 items), treatment and management (2 items), protective factors (5 items). The score was calculated by summing the correct answers for each item, yielding a total score ranging from 0 to 27 points. A higher total score indicated a better knowledge about dementia. The scale showed adequate internal consistency, with a Cronbach’s alpha coefficient of 0.782 for the total scale. 

Sociodemographic questions assessed potential moderators of dementia knowledge. The questionnaire included a set of demographic and background questions about gender, age, living location (urban or rural), and years of education. In addition to sociodemographic features, personal experience of exposure to information about dementia also has a certain impact on knowledge about dementia and attitudes towards people with dementia [16]. Thus, we also included the factor ‘‘experience of exposure to dementia’’ in our analyses. Experience of exposure to dementia was represented using two items: “Have you heard of dementia or dementia patients (yes/no)?” and “Have you been in contact with dementia patients (yes/no)?”

SO JUMP, like SurveyMonkey, is a professional online questionnaire survey, evaluation, and voting platform in China. We used SO JUMP to create a questionnaire comprising knowledge about dementia and AD. According to the 43rd statistical report on the development of the Internet in China [36] till December 2018, the number of Internet users was 829 million, among which the number of mobile Internet users was 817 million, constituting 98.6%. WeChat is a simple Chinese mobile applet, like WhatsApp in some countries, widely used and convenient, that does not require special computer skills [37]. There are 780 million WeChat users in China, accounting for 95.5% of mobile Internet users, with a WeChat friend circle usage rate of 83.4% [36]. The WeChat-based approach has been widely applied in questionnaire collection [38], health education and management [39,40], and intervention studies [41]. So, this study conducted a survey on the public knowledge of dementia through WeChat friend circles.

In order to avoid being treated as a nuisance or a virus by direct releasing, we recruited sophomores as the seed investigators and released this survey via their WeChat friend circles. The participants completed the survey presented on SO JUMP via the WeChat platform. No identifying information was collected for any participants. With the help of WeChat, the overall sample was randomly chosen. 

### 2.3. Data Analysis

Descriptive statistics were used to summarize participant sociodemographic characteristics, including the variables of gender, age, years of education, living location (urban or rural), and experience of exposure to dementia (heard of dementia/dementia patients or been in contact with dementia patients). Median and Interquartile Range (IQR) and percentage were chosen to describe the sociodemographic data and knowledge scores. The Mann–Whitney U test and the Kruskal–Wallis test were used to compare the knowledge scores between subgroups based on participants’ characteristics. The Kolmogorov–Smirnov test was conducted to determine the normality of distribution of the mean knowledge scores before comparing the scores. Non-parametric tests (Mann–Whitney U and Kruskal–Wallis tests) were used to compare mean scores between subgroups, because the knowledge scores were not normally distributed (*p* < 0.001) [19]. The statistical analyses were undertaken using IBM SPSS Statistics for Windows (Version 24.0., IBM Crop., Armonk, NY, USA). Statistical significance was set at *p* < 0.05 with two-side tests.

## 3. Results

### 3.1. Demographical Characteristics of Respondents

A comprehensive study of 9800 Chinese participants, which included 5100 women (52.04%) and 4700 men (47.96%) was conducted. Most of the participants (3940, 40.20%) were aged 18–20 years, and a small fraction of the sample was aged 60 or above (1324, 13.51%). According to the distribution by education level, 54.12% of the individuals had college education, and 10.61% had primary school education or lesser. Participants living in rural areas accounted for 48%, and the rest were living in cities. About the experience of exposure to dementia, 94.66% respondents replied that they had heard of dementia, and 42.41% participants answered that they had had contact with dementia patients at least once. Table 1 presents the statistics for other individual characteristics.

### 3.2. Knowledge about Dementia

On average, the participants answered 63.14% of the questions correctly. There was significant variability in the level of knowledge of dementia, and the percentage of correct responses to each item ranged widely from 28.67% (lowest) to 90.58% (highest). Regarding the symptoms of dementia, memory loss was identified with the highest accuracy (90.58%); the second highest score was obtained for the question about the possibility of preventing a person from developing AD, with 87.57% of correct answers. The lowest correct answer rates were 28.67% and 29.85% for the questions about high cholesterol and diabetes increasing a person’s risk of developing AD, respectively. The percentages of overall correct answers for the category items “protective factors” and “treatment and management” of dementia were relatively high, with mean percentages of correct answers of 77.01% and 71.99%, respectively. However, the questions regarding “risk factors” of dementia obtained the lowest percentage of correct answers (50.84%) (Table 2).

### 3.3. Association between Socio-Demographic Characteristics, Experience of Exposure to Dementia, and Knowledge about Dementia

The dementia knowledge median score was 17 (14, 20) points out of 27 points. There were statistically significant differences in dementia knowledge scores between males (16 (14, 20) points) and females (17 (14, 20) points, *p* =< 0.001). The scores prominently increased with age for the participants aged 18 to 60 years (*p* < 0.001), but not for the group aged 60 years or above. The total scores of dementia knowledge were significant higher for the subgroups with a higher education level than for the subgroups with a lower education level (*p* < 0.001). Sure enough, the dementia knowledge scores varied with the experience of exposure to dementia. First, the dementia knowledge score of the participants who answered they had heard of dementia was 17 (14, 20) points, whereas that o those who answered that they had not heard of dementia was 14 (12, 18) points (*p* < 0.001). Second, the knowledge about dementia among respondents who had had contact with dementia patients (17 (14, 21)) was obviously higher than that among respondents who had had no contact with dementia patients (16 (14, 19); *p* < 0.001). Additionally, dementia knowledge was statistically different between rural residents (16 (13, 20) points) and city residents (17 (14, 21) points, *p* < 0.001; Table 3).

## 4. Discussion

For years, pen-and-paper questionnaires have been a trusted and validated data collection method [38]. However, it is not easy to carry out large-scale surveys with a wide geographic coverage of the target population if human, material, and financial resources are limited. With the development of the Internet, network surveys have played an increasingly important role in scientific research and clinical practice, and the validity and reliability of questionnaires administered online have been demonstrated [42]. Our study, based on mobile internet and the WeChat friend circles of clinical medicine students has several advantages in terms of the sample: (1) Thanks to the popularity of the mobile network and WeChat friends circles, the survey could be carried out smoothly; (2) Our undergraduate seed investigators came from all over the country. The WeChat online survey allowed unprecedented investigator access to large samples of demographically diverse respondents from across the whole country; (3) Every sophomore over the age of 18 has a We-chat account, and they are the main group of mobile Internet users; (4) The standard survey procedure is a necessary skill for medical students, so they took the survey seriously; (5) WeChat online surveys can save costs, in terms of both time and resources, and avoid errors and omissions in data entry. This has been confirmed by Sun et al. [38]; (6). The participants completed the online survey more easily and enjoyably than with a face-to-face survey [38], and researchers were able to obtain a superior response rate. The number of our responders was 10,562; (7) WeChat online surveys can help avoid the awkwardness of face-to-face surveys, and the results can more truthfully reflect the real situation of the respondents.

Our study found that the population’s total dementia knowledge rate was 63.14%, which was an overall poor level compared to those found in other studies [10,16,19]. Considering that the predominant group was a group of highly educated (≥13 years of education, 54.12%) and young people (≤40 years, 65.51%), the actual awareness rate of the public would be even lower. From the knowledge category, the identification rate of risk factors was the lowest (50.84%), meaning that half of the participants could not recognize the risk factors correctly. Also, 79.09% of people believed that the likelihood of having AD increased with age, but only 54.53% were able to recognize that family AD history was associated with AD. Family history of AD is not necessary for an individual to develop the disease; however, individuals who have a parent, brother, or sister with AD are more likely to develop AD than those who do not have a relative with AD [43]. More than half of the participants recognized brain damage (66.18%) and stroke (64.31%) as risk factors for developing AD. That means many people continue to confuse brain injury problems with AD. Other factors, such as hyperlipidemia (28.67%), diabetes (29.85%), and hypertension (42.06%), were deemed as low-risk factors of dementia. In reality, growing evidence suggests that obesity in midlife [44], diabetes [45,46], midlife hypertension [44,47], and midlife high cholesterol [48,49] are closely implicated as risk factors for dementia. Whereas age and family history cannot be changed, other risk factors can be modified. Understanding these modifiable risk factors for dementia may encourage preventative health behaviors in youth and mid-life adults, ultimately reducing the late-life incidence risk of cognitive decline and dementia [50]. Although hypertension and diabetes are included in standard chronic disease management in China, there is still a long way to go to reduce the risk of dementia by achieving control over high-risk cardiovascular diseases. 

We found that 90.58% of participants could answer correctly that patients with AD have a gradually worsening ability to remember new information (Table 2). Therefore, although the recognition of amnesia as a symptom of dementia may be good, public knowledge may not allow an easily detection, leading to the screening and diagnosis, of non-amnestic dementia. The acceptance level of AD patients’ abnormal behaviors and emotional changes was relatively low, but depression or social isolation and dementia are mostly bidirectional [51]. Therefore, due to lack of recognition, dementia may worsen. Werner et al. [52] found that a higher level of knowledge about AD symptoms was likely to lead to timely dementia diagnosis and initiation of continuous treatment. Therefore, we need to make an effort to increase the knowledge of dementia symptoms.

The mental and psychological burden on family members due to dementia in relatives was deemed as having the largest impact (86.92%), which was higher than the effects on work and study (72.60%) and economic burden (68.12%). The word “dementia” is derived from the Latin words de (out of) and mens (mind), indicating that dementia was considered as a punishment or a curse, leading to the stigmatization of cultural beliefs about dementia [26,53]. Almost one-third of the participants believed that social discrimination against AD patients existed [26]; this kind of misunderstanding and public discrimination may increase the psychological burden for the relatives and caregivers of AD patients [26]. Compared with non-caregivers of similar ages, approximately 40% to 59% of family caregivers of AD patients rated the emotional stress or depression of caregiving as high or very high [54,55]. AD family caregivers had to work fewer hours or stop working entirely to support the patients. Almost 54% of dementia family caregivers who were employed had to go to work late or leave early, 15% had to take a leave of absence, and 9% ultimately quit their jobs to continue providing care [43]. The total cost of dementia, including the direct, indirect medical payments, and other non-medical payments, is significantly higher than predicted. The direct medical payment can be partly covered by medical insurance, but the cost of AD health care, long-term care, and hospice services may place a substantial financial burden on families [43]. To care for dementia patients, families often have to take money out of their retirement pensions, cut back on buying food, and reduce their own trips to the doctor [43]. People with dementia still incur high out-of-pocket costs. 

Most participants thought that physical exercise (86.36%), social activities (83.16%), healthy lifestyles (81.33%), and cognitive training (76.68%) are helpful to prevent dementia. The Chinese participants in the present study were more likely to identify protective factors for dementia than the Irish participants in a similar study [17]. 

Although nearly half of the participants (56.42%) reported that currently AD patients are hardly cured, similar to results found in South Korea [19], the vast majority (87.57%) believe that AD could be prevented. That means that, although people expressed a negative perception toward dementia treatment, they held positive perceptions toward dementia prevention and management. The current lack of confidence in AD treatment may be related to the negative results of several Food and Drug Administration (FDA) research studies [43]. Although there is a lack of disease-modifying therapies for AD, people think it is possible to improve the patients’ quality of life through an active management of dementia [56].

There is no general pattern indicating that either male or female participants had better dementia knowledge. Several studies found no or only little gender-based differences in knowledge of dementia [13,28,57]. This survey showed that women had a higher level of knowledge about dementia than men. The fact is that women, especially middle-aged Chinese women, are often responsible for taking care of their families, including old parents or someone with dementia. Approximately two-thirds of AD caregivers are women [58]. More specifically, they are daughters, wives, or nurses [59]. Female caregivers are more likely to be concerned with someone with problems. In addition, some females might be aware that gender is one of the risk factors for AD [26], which could increase their attention.

In terms of age, for people aged 18–60 years, there was a clear increase in the dementia knowledge score as the age of the participants increased, but the group aged over 60 years showed a dementia knowledge similar to that of the group aged 18–20 years. This result suggests that young and middle-aged people (20–60 years old), known as the “sandwich generation,” are more concerned about dementia than young (18–20 years old) and old people (≥60 years old). In China, most AD patients are cared for by their families, and the sandwich generation is the mainstay of family care, taking care not only of the children, but also of aging parents, including dementia patients. In contrast, only 23% of dementia caregivers constitute the sandwich generation in the US [43]. The Medicare system of China only covers a small amount of treatment and nursing expenses for AD patients, causing a further economic burden for the families of dementia patients, particularly for the sandwich generation [60]. The biggest worry is that people over 60 years of age, at the highest risk of dementia, are less concerned about dementia knowledge than others. There may be two reasons for this phenomenon. First, because of the fear of being labeled as dementia patients, they are more likely to believe that various symptoms are normal manifestations of aging rather than of AD [61]. Second, they are retirees and cannot easily access information on dementia through more or better formal channels in China. Therefore, the government, society, and family should work together in popularizing dementia-related knowledge. 

The relationship between education level and dementia knowledge level found in this study is consistent with that reported in previous studies [18,19,24,62]. People living in urban areas had a higher knowledge level of dementia than those living in rural areas. However, the following situations have to be considered: (1) More than 70% of Chinese older adults live in rural areas [63,64], and the prevalence of dementia is approximately 5.6% for rural towns in China [64]; (2) Rural residents are relatively less educated, and there is still a large number of people with no education or primary school education [63]; (3) In fact, most physicians cluster in cities, and relatively few neuropsychiatric professionals work in rural towns, indicating that many older adults with dementia cannot obtain professional diagnosis and treatment; (4) A more serious fact is that people with early-onset dementia are often referred back to their communities for follow-up care, because of a lack of medical insurance and limited medical resources in rural China [65,66]. Therefore, given the situation of rural residents at a high risk for and with low awareness of dementia, we must recognize the importance of educating about dementia to allow for prevention and control.

Experience of exposure to information on dementia had a great impact on participants’ dementia knowledge. Approximately 94.66% of the participants said they had heard of dementia, and their awareness degree was significantly higher than that of those who had not heard of it. Although only 42.41% of the participants had had contact with dementia patients, their dementia knowledge scores were higher than those of participants who had not had any contact. It has been suggested that exposure to dementia is helpful to improve the public understanding of dementia in previous studies [16,28,67]. This is a positive finding. The participants with more exposure experience of dementia have more positive attitudes towards dementia. They have less prejudice and are more likely to correctly recognize some abnormal behaviors or behaviors of dementia. They tend to be less skeptical about early detection of dementia and want to be screened for AD even on a regular basis [68]. Simultaneously, we should also be vigilant about the correctness of AD information, because non-professional information is more likely to bring fear of AD [26]. Zeng et al. found that most people thought that they were not well informed about AD through public education by the government, media, or medical institutions [26]. It is urgent to pursue educational programs and campaigns on dementia, especially through media and publications, with China’s governmental support [12].

This study has some limitations. First, the modified dementia knowledge scale was revised on the basis of the Chinese version of ADKS. More studies are needed to verify whether it can be widely used. Second, the survey was based on WeChat, and may have missed some participants who were not WeChat users. Third, the respondents were mostly highly educated young people from cities and towns, with a smaller number of less educated old people from rural areas, because we used a WeChat survey. Although the survey participants’ crowd structure is consistent with that of WeChat users in China, it is differences from that of the general Chinese population. Therefore, the level and characteristics of dementia knowledge of the Chinese population might be different from those we determined, and we should be cautious in popularizing these results.

## 5. Conclusions

This survey found that Chinese people have a lower level of knowledge about dementia than people in high-income countries, especially in terms of risk factor identification. The lack of understanding of high-risk factors of dementia (hypertension, hyperlipidemia, and hyperglycemia) naturally leads to insufficient prevention and control of dementia. In addition, the group at higher risk for dementia (people older than 60 years, with low education, living in rural areas), demonstrated a lower the level of dementia knowledge. This current situation of limited knowledge about dementia is likely to pose many challenges for the prevention and treatment of dementia in China. The experience of dementia exposure, meaning having heard of or been in contact with dementia patients, will help people better understand dementia and dementia patients. We suggest that further educational programs and campaigns should be implemented to improve the knowledge of dementia in China, with greater focus on the older population as a target audience and emphasizing the risk factors of dementia as educational content.

## Figures and Tables

**Table 1 ijerph-16-04231-t001:** Sociodemographic characteristics (N = 9800). AD: Alzheimer’s disease.

	Characteristics	Participant Number (N)	Percentage (%)
Sex	Male	4700	48.00
	Female	5100	52.00
Age (years)	18–20	3940	40.20
	21–40	2480	25.31
	41–60	2056	20.98
	60~	1324	13.51
Education level (years)	0–6	1040	10.61
	7–9	1444	14.73
	10–12	2012	20.53
	≥13	5304	54.12
Resident	Rural	4704	48.00
	City	5096	52.00
Experience of exposure to dementia			
Heard of dementia/dementia patients	Yes	9277	94.66
	No	5,23	5.34
Contact with AD patients	Yes	4156	42.41
	No	5644	57.59

**Table 2 ijerph-16-04231-t002:** The percentage of correct answers for questions regarding the knowledge about dementia. T: true, F: false.

Modified Dementia Knowledge Scale (Correct Answer)	Category	Percent Correct (%)	Average Correct Rate (%)
	total		63.14
Many people with AD have trouble in handling money or paying bills. (T)	Symptoms	56.06	62.64
Some people with AD may become lost even in a familiar environment. (T)		78.98	
Most people with AD remember recent events better than things that happened in the past. (T)		90.58	
Most people with AD cannot name persons or things familiar to them. (T)		79.66	
Many people with AD may be at a loss when they are in trouble. (T)		53.35	
Some people with AD may wear more clothes in summer and less in winter. (T)		41.31	
Some people with AD do not care about surrounding things. (T)		47.16	
Some people with AD are more willing to sit at home, not go out, and not communicate with others. (T)		54.07	
If someone’s biological parents have Alzheimer’s disease, this person is more likely to get Alzheimer’s disease too. (T)	Risk factors	54.53	50.84
The chance of developing dementia will increase as you get older. (T)		79.09	
Having high blood pressure may increase a person’s risk of developing AD. (T)		42.06	
Having high cholesterol may increase a person’s risk of developing AD. (T)		28.67	
Diabetes may increase a person’s risk of developing AD. (T)		29.85	
Brain trauma may not increase your chances of suffering from AD. (F)		66.18	
Stroke or brain surgery have no effect on the chances of developing AD. (F)		64.31	
Chronic systemic diseases do not increase the chances of developing AD. (F)		42.04	
A person’s work or study may get affected when a family member is experiencing AD. (T)	Life impact	72.60	66.97
The likelihood of family members getting other diseases will increase when some people get AD. (T)		40.90	
A person could experience a significant financial burden when a family member gets dementia. (T)		68.12	
The families’ psychological burden and the occurrence of psychological illness may increase when a person is diagnosed with AD.(T)		86.29	
Alzheimer’s disease cannot be cured. (T)	Treatment and management	56.42	71.99
It is impossible to prevent a person from developing AD. (F)		87.57	
Maintaining a healthy lifestyle may reduce the risk of developing AD (T)	Protective factors	81.33	77.01
Physical exercises are generally beneficial for people experiencing AD. (T)		86.26	
Intellectual activities such as reading books or newspaper and playing chess or card games are helpful for people with AD. (T)		76.68	
Active participation in various social activities may protect the cognitive function of AD patients. (T)		83.16	
Living with family may help someone with AD to preserve their cognition. (T)		57.63	

**Table 3 ijerph-16-04231-t003:** Relationship between socio-demographic characteristics, experience of exposure to dementia, and knowledge about dementia.

		Total Score (out of 27) Median (p25, p75)	Z (or H)	*p*-Value	Z#	*p*_1_-Value
Total score		17.00 (14, 20)				
Sex	Male	16.00 (14, 20)	−6.31	<0.001 *		
	Female	17.00 (14, 20)				
Age (years)	18–20	16.00 (14, 19)	410.77	<0.001 *		
	21–40	18.00 (15, 21)			−14.34	<0.001 *
	41–60	18.00 (15, 21)			−15.62	<0.001 *
	60~	15.00 (13, 19)			1.72	0.513
Education level (years)	0–6	15.00 (12, 19)	95.55	<0.001 *		
	7–9	17.00 (13, 20)			−4.83	<0.001 *
	10–12	17.00 (14, 21)			−9.39	<0.001 *
	≥13	17.00 (14, 20)			−8.03	<0.001 *
Resident	Rural	16.00 (13, 20)	−10.24	<0.001 *		
	City	17.00 (14, 20)				
Experience of exposure to dementia						
Heard of dementia	Yes	17.00 (14, 20)	−12.55	<0.001 *		
	No	14.00 (12, 18)				
Contact with AD patients	Yes	17.00 (14, 21)	−9.18	<0.001 *		
	No	16.00 (14, 19)				

# represents the Z value of each subgroup compared with the first group; p_1_ means the first subgroup is the control.

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
