# Peer review of "Public Knowledge about Dementia in China: A National WeChat-Based Survey"

_ijerph, 2019, doi:10.3390/ijerph16214231_

Round 1

Reviewer 1 Report

The paper is of high interest regarding the evaluation of dementia awareness and information needs in China.

There are some issues which have to be processed:

PAGE 0 - Lines 42-43:

I am confused about the annotation that "65% of AD cases worldwide are in China"

China has about 10 million people with dementia and worldwide there are about 47 million people with dementia. AD is about 80 percent of all dementia forms…

See also, ADI-Report - Prince, M. J. (2015). World Alzheimer Report 2015: the global impact of dementia: an analysis of prevalence, incidence, cost and trends. Alzheimer's Disease International.

PAGE 2 - Lines 106 - 112:

This paragraph already belongs to the methods section in my understanding.

PAGE 2 - Lines 115 - 119:

Why have been only medical students recruited in the first place? In the “limitation” section it was pointed out that: "In addition, we tend to recruit more highly educated young people from cities and towns, while ignoring most of the less-educated old people in rural areas." (P9 - L380)

Has the one aspect something to do with the other one? Why have you chosen this process of generating a potential "selection bias"? This was one aspect, you criticized related to the former conducted Chinese studies to this topic within the introduction.

PAGE 2 - Lines 124 - 125:

The ethical approval document number is not mentioned.

PAGE 2 - Lines 133 - 134:

Is there a Chinese version of the questionnaire existing? If not, how was it translated into Chinese language? If there was no professional translation and re-validation of the instrument, it has to be mentioned in the "limitations" section.

PAGE 5 - Lines 201 - 202:

The age span is very high in group 1 (18 to 60 years). In the discussion section, another age span group (18-20) was mentioned. It would be very interesting to compare also different age groups, e.g. between 30 and 50 years, for example. 

PAGE 10 - Line 410

The answer you have given related to potential "conflicts of interest" is not an appropriate answer of the question. 

GENERAL:

Some passages of the Introduction seem a bit too extensive argued (e.g. Lines 62-70; page 1). Maybe, shortening is possible.

Reviewer 2 Report

It is a large study and the authors dealt with the data in more robust ways. My only concern is as the data collected in the Chinese language, how data were translated into English and by whom? I have noticed that the authors mentioned the study limitation. I think a separate sub-heading for limitation would be more suitable?

Reviewer 3 Report

Thank you for the opportunity to review this paper which presents an interesting study of dementia knowledge in a broad cross section of participants.

Both the introduction and Discussion could be considerably reduced in length. Studies referred to on page 2 - lines 70-96 should be framed in terms of their results rather than listing study locations and populations alone. The introduction does not really present what is currently known about knowledge of dementia in the global setting and the Chinese setting particularly. Possibly less time could be spent on justifying design in the introduction- this may be more appropriately moved to methods.

Line 114- on page 3- Particpants. It was not clear exactly ehat the role of seed investigators was. It was not clear how the survey was conducted- was it on line, was it self administered or did it involve interview.  The methods indicates verbal consent was sought- it is unclear how this process occurred.

Line 129 page 2- the ADKS has been widely used, but a number of other knowledge assessment tools exist- the statement that it reflects the latest research evidence is unsupported.

Given there was substantial adaption and possibly translation of the Tool, it should not be referred to as the ADKS particularly  as the authors indicate they made a comprehensive adjustment to the questionnaire's contents. Changing the scoring approach may alter the described psychometric properties of the tool. In reviewing the questions in Table 2- the meaning of the questions may not have translated properly from Chinese. Two questions in particular are of concern- In rare cases people have recovered from AD- I find it most surprising that the answer is indicated as True and It has been scientifically proven that mental exercise can prevent a person from getting AD is also indicated as true- both of these should be false. The first questions which use the term "most" might be better translated as "some" or possibly "many". People with AD do not care about surrounding things- does not translate well as this may be true of some people but could not be considered true for all.

Line 201 page 6- "distinct higher" should be significantly.

The use of the term "demented" is not advised.

The discussion could be considerably shortened and focussed on knowledge issues- there are multiple speculative statements which ar enot substantiated from the data presented her.

I draw the authors attention to the following lines which include these statements or terms that are not familiar to many readers line 228, 230-233.

Line 238 you refer to a global rate but do not provide a reference. .

On page 8   lines 264-265- you indicate that - most common symptom was the worsening ability to remember new information- this was not confirmed by this study.

Speculation line 272-276- is not substantiated not is line 298-302 and are not conclusions that can be drawn from this study

Reference 59 is incorrectly cited.

Non Chinese readers may not be familiar with SOJUMP

Line 165 pge 4- you report the data as Mean +/- SD but indicate the data is not normally distributed- Median and IQR may be more appropriate.

In table 2- you indicate an overall correct rate of 63.14%- it is unclear whether this is the proportion of respondents who achieved a perfect score or if this is the average score ( which I believe it is). A better way to present this would be to say that on average participants  answered 63.14% of the questions correctly. Line 189 uses the term rightness - please review this section for correct use of english - you could consider saying 87.5% of respondents correctly answered.

On page 9/10 the terms confidence and agreement and attitude seem to be used interchangably-these are very different constructs and care should be taken in determining whether a correct answer reflects confidence. Lines 335-337 is speculation.

Round 2

Reviewer 1 Report

Parts of the process of the participants recruitment (medical students) is still a weakness but this aspect is noted in the limitation section appropriate.

In conclusion, it is an interesting, well written paper with high relevance and novelty regarding the Chinese setting.

Best regards

Reviewer 3 Report

Thank you for the opportunity to re- review the manuscript. I still have a concern regarding two items in the modified ADKS,

The original items from ADKS are false and the text indicates that the authors have calculated them as true. Two things are important- the correct  translation from Chinese and whether the calculation is correct- the changes the authors have made in text must reflect the actual question asked in Chinese- care must be taken that this is accurate. I could not find reference to these two statements in the world Alzheimer report. I apologise if I have made an error.

The following site provides the correct questions and answers- including evidence of the translation mechanism to confirm accuracy may assist.

https://pages.wustl.edu/files/pages/imce/geropsychology/adks_with_and_without_correct_answers.pdf

2. It has been scientifically proven that
mental exercise can prevent a person
from getting Alzheimer’s disease.
(FALSE)

8. In rare cases, people have
recovered from Alzheimer’s disease.
(FALSE)

Until these are corrected the overall calculations are incorrect.

Other than this the authors have addressed most of the issues, although I would suggest that the term "demented" is inappropriate
